# Spam Reviews Detection in the Time of COVID-19 Pandemic: Background, Definitions, Methods and Literature Analysis

**Ala' M. Al-Zoubi** [1,*] , **Antonio M. Mora** [2] **and Hossam Faris** [3,4,5]

1   School of Science, Technology and Engineering, University of Granada, 18010 Granada, Spain
2   Department of Signal Theory, Telematics and Communications, ETSIIT and CITIC, University of Granada, 18071 Granada, Spain; amorag@ugr.es
3   King Abdullah II School for Information Technology, The University of Jordan, Amman 11942, Jordan; hossam.faris@ju.edu.jo
4   School of Computing and Informatics, Al Hussein Technical University, Amman 11831, Jordan
5   Research Centre for Information and Communications Technologies of the University of Granada (CITIC-UGR), University of Granada, 18010 Granada, Spain
*   Correspondence: ala.m.zoubi@gmail.com or alzoubi@correo.ugr.es

**Abstract:** During the recent COVID-19 pandemic, people were forced to stay at home to protect their own and others' lives. As a result, remote technology is being considered more in all aspects of life. One important example of this is online reviews, where the number of reviews increased promptly in the last two years according to Statista and Rize reports. People started to depend more on these reviews as a result of the mandatory physical distance employed in all countries. With no one speaking to about products and services feedback. Reading and posting online reviews becomes an important part of discussion and decision-making, especially for individuals and organizations. However, the growth of online reviews usage also provoked an increase in spam reviews. Spam reviews can be identified as fraud, malicious and fake reviews written for the purpose of profit or publicity. A number of spam detection methods have been proposed to solve this problem. As part of this study, we outline the concepts and detection methods of spam reviews, along with their implications in the environment of online reviews. The study addresses all the spam reviews detection studies for the years 2020 and 2021. In other words, we analyze and examine all works presented during the COVID-19 situation. Then, highlight the differences between the works before and after the pandemic in terms of reviews behavior and research findings. Furthermore, nine different detection approaches have been classified in order to investigate their specific advantages, limitations, and ways to improve their performance. Additionally, a literature analysis, discussion, and future directions were also presented.

**Keywords:** online reviews; spam reviews; detection; COVID-19; survey

## 1. Introduction

In the past couple of years, the world has been somehow on hold due to the COVID-19 pandemic. Due to this, remote tech usage reached a climax in all areas, including, education [1], health care [2], working [3], and shopping [4,5]. For each, a discussion was required in order to choose the most appropriate choice, especially when it came to online shopping. Therefore, online reviews have become an important part of a customer's decision-making process. Such reviews are used progressively by organizations and individuals to perform business and purchase verdicts. Reviews of products and services are generated by users' experiences, and they have a significant impact on customers' purchase decisions [6,7].

Consumers can, therefore, obtain some information about the product or service they are considering purchasing by reading a few reviews. Also, organizations can modify their products or redesign their business strategies according to these reviews [8]. For instance, a number of consumers buy a particular model of laptop, and then they post a review

regarding the quality of the internal keyboard. In order to meet consumers' expectations, the manufacturer can consider these reviews and adjust the keystrokes. Thus, positive reviews can cause major financial gains, while negative reviews may even lead to sales loss [9]. As a result, more merchants are now taking into account the general public's opinions for the products' decision making [10–12]. For example, in 2004, one consumer wrote a review about the U-shaped Kryptonite lock and how it can easily be opened using a ballpoint pen [13]. This hustle caused by this review forced the company to replace any affected lock without any charge. The Kryptonite incident attests to the importance of online reviews.

Furthermore, due to the detrimental situation (COVID-19 quarantine), reading and posting reviews across all platforms have increased rapidly [14]. Unfortunately, the rise affected not only legitimate reviews but also spam reviews. Spam reviews are reviews characterized as fraud, fake, malicious, and false opinions intended for publicity or profit. Due to this attitude, most consumers and organizations would be misled away from making the correct decisions [15]. As a result of spam reviews, it became harder to improve products and services because of the failure to recognize feedback and reviews from real customers.

This kind of review can be easily posted without any constraint on several platforms, so certain product providers or vendors could abuse this to promote their products and services or to disparage their competitors. Media news, such as BBC and New York Times, reported that there are more spam reviews on websites than real ones, for example, a photography company revealed to have posted thousands of spam reviews [16]. Accordingly, detecting spam reviews is a requisite nowadays, since without solving this issue, consumers will stop trusting completely online review platforms [17].

To counter this issue, industry and academia expand their efforts to identify spam reviews in all possible ways. In the literature, there are many purposed approaches tackle the problem of spam reviews detection recently, such as supervised learning [18–22], integrated with sentiment analysis method [23–27], graph based detection [28,29], ensemble [30], semi-supervised learning [31–33], deep learning [34,35] and others methods like [36–38].

However, outlining and analyzing all these works and more can be bothersome for others. Therefore, several survey papers were proposed in order to examine, investigate and address different works regarding spam reviews detection. For example, Ref. [15] was one of the first surveys that examined the spam reviews detection works. The survey systematically analyzed and categorized various models of spam reviews detection as well as evaluated them in terms of accuracy and other measures. In the same year, the authors of [39] also introduced a survey that handles the problem of spam reviews detection. Nevertheless, they only considered works that related to machine learning techniques. They stated that most approaches, at that time, focused on supervised learning methods. Furthermore, Rajamohana et al. proposed a study that exchange views of different frameworks adapted for detecting spam reviews [40]. Their paper provided a comprehensive comparison of accuracy levels that tried to solve spam reviews detection. Whereas, Aslam et al. presented a survey focused on various machine learning techniques and compare their accuracies for the purpose of identifying spam reviews [41]. Their conclusion suggested that the semi-supervised techniques combined with multi-aspect features have the best performance in detecting spam reviews.

As for [8], they summarized and outlined various spam reviews datasets alongside their implemented methods. Also, they categorized the methods into two parts, the neural network model and traditional statistical methods. While [42] conducted a comprehensive review of the spam reviews detection works through utilizing the Systematic Literature Review (SLR) mechanism. More than 70 papers have been reviewed and examined, and then they study the feature extraction techniques in these works. Besides, additional analysis has been preform regarding the metrics, datasets, and performance of the detection approaches. The authors of [43], presented a literature review of the works of spam reviews detection from 2015 to 2020. They identify all studies during that time and scrutinize the findings, research gaps, and similarities of the approaches. Additionally, a three par-

tition was applied to the analyzed works based on who write spam review, detection methods, and spammer groups. Likewise, Wu et al. proposed a survey according to the antecedent–consequence–intervention technique to investigate spam reviews [44]. Nearly 20 and 18 research questions and propositions, respectively, were determined after examining the literature. Further, due to the lack of excellent datasets, the works on spam reviews did not expand as expected, therefore, a comprehensive outline of the existing public datasets has been presented.

On the other hand, Ref. [45] investigated the ground truth of spam reviews detection studies. This review focused on state-of-the-art truth for two classical aspects, expert spammers and crowdsourcing. After their analysis, four conclusions have been reported. Firstly, data collection was more challenging if the spammers are professional. Secondly, the behavioral type was more reliable than the linguistic type. As for the third conclusion, the abnormal activities can be dependable at the same level as spam intentions. While in the fourth point, several reliable facts have been identified, namely, spam cost, deviation, opinion proportion, grouped spamming, review distribution, and grouped spamming. Rodrigues et al. proposed a different survey that focused on spam reviews detection combined with the sentiment analysis method [46]. Moreover, the authors of [47], provided a review of the existing strategies and methods for detecting spam reviews. They performed a taxonomy on the methods related to machine learning techniques.

Regarding the most recent surveys, three works have been proposed this year. Starting with an investigation of the spam group detection works addressed by [48]. In this study, the authors suggested -after reviewing a number of papers- that the graph-based detection methods show better performance than the other approaches. Also, the behavioral features obtained the best detection accuracy among other features. Whereas, Mohawesh et al. summarized the available datasets and analyzed the feature extraction techniques of the existing approaches [49]. Also, the work-study and compared the standard machine learning techniques against the deep learning methods performance in detecting spam reviews. Further, current gaps and a future direction of the research domain have been discussed. Paul and Nikolaev [50] presented a novel survey that covers the commercial and basic research solutions. Also, the reasons behind the limitation of current approaches for the detection methods.

The previously mentioned surveys covered various aspects of spam reviews detection works, ranging from, performance comparison of specific techniques, feature extraction used, measures, available datasets, analysis and suggestion of best methods, approaches limitation, to the future works of the research area.

In order to identify relevant articles for this survey, selection criteria were required. The selection criteria comprise inclusion and exclusion phases. If a duplicate article showed up in the inclusion list more than once from different sources, it will still be included only once. For the selection of related articles, inclusion criteria were used sequentially as follows:

- Articles whose titles are related to some or all of the search keywords.
- Articles that contain keywords that are subsets of the search keywords.
- Articles that their abstract portray spam reviews detection (or its synonym).
- Articles that suggest new models or techniques for spam reviews detection or alter an existing one.
- Articles that used already-existing spam reviews detection methods in their phase of the experiment.

    While the exclusion criteria for removing irrelevant articles follow the following points:

- Articles published before 2020 or those not published during the COVID-19 pandemic.
- Articles that do not have or include experiments on spam reviews detection.
- Articles that do not meet any of the inclusion criteria.

    This up-to-date survey differentiates from the previous surveys in consecrating on the works that were proposed during the COVID-19 situation. And also due to the rapid

alteration and increase of spam reviews in this period. In addition, a detailed background of the related concepts has been introduced including, online reviews, spam reviews, detection methods, and datasets. Categorize the detection methods into several groups (see Figure 1) and analyze the mechanism used for each category alongside their limitation, strength and provide solutions to enhance their performance.

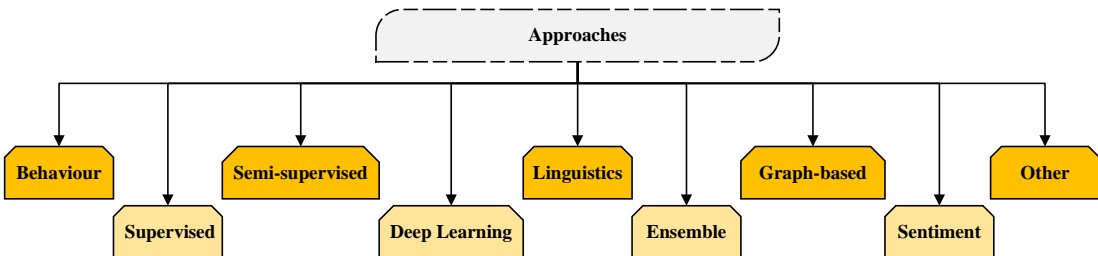

**Figure 1.** Type of approaches to detect spam reviews used in this study.

Also, this survey identifies the differences, development, and outcome of the spam reviews works prior to and during the COVID-19 pandemic, namely:

- The increase of reviews amounts due to the rise of people staying inside.
- The behavior of reviews differentiates between the two periods in products, services, and consumers' target, style of context, and their important impact.
- Spam reviews also show growth in numbers thanks to the increase of individuals reading and searching for reviews.
- Detection technique models improve and become more varied as can be seen in Section 3.

Therefore, the contribution of this work is summarized in the following points:

- Outline the previous spam detection surveys and their contributions.
- Provide a background investigation about the concepts of spam reviews detection environment and specify their definitions and motivations.
- Address the works and studies during the COVID-19 situation (2020 & 2021) in order to analyze the detection methods and reviews type at that time.
- Analyze all works and address their limitations, advantages, and how to improve them.
- Classify the detection methods into nine different categories.
- Present a literature analysis, discussion, and future directions of the spam reviews detection.

The remainder of the paper is divided as follows: Section 2 presents a detailed background of the used spam reviews concepts. Section 3 proposed a review about the detection methods and approaches for spam reviews alongside their definitions. The literature analyses, discussion, and future directions are introduced in Section 4. Finally, the conclusion of this paper is provided in Section 5.

## 2. Background, Definitions and Motivation

### 2.1. Online Reviews

A review can be defined as feedback of specific products from specific users to inform particular individuals. De Pelsmacker et al. [51] described online reviews as Electronic word-of-mouth (eWOM). The eWOM is informal communications about usage or features of goods, services, or their sellers geared towards consumers through an internet-based technology. On the other hand, Ref. [52] stated that reviews are popular user-generated content (UGC) that consumers utilize when doing different decisions related to travel, buying, renting, and using a service. Reading such reviews can reduce risk, make new ideas, streamline choices and emphasize choices for consumers. While the work in [53], interprets online reviews as a process that previous consumers provide to criticize products.

That being said, reviews are known as an information source for marketers and consumers in order to learn product quality. Also, reviews are considered a useful metric

to measure the loyalty of consumers in situations like product's success critical implications [54]. Therefore, providing reviews can benefit many related parties to study the products in different aspects to confirm the quality, sales, reputation, and market share of the product. Others consider reviews more credible and essential than the data presented by the commercial sources [55].

Moreover, reviews could affect several facets of products such as revenue, prices, performance, and popularity. According to European Consumer Centres, there are more than 80% of consumers read reviews before purchasing [56]. Online reviews have a primary and critical impact on e-commerce, particularly, shopping decisions and the amount of money that could be spent. For that reason, reviews are known as one of the important aspects of business performance [57].

Reviews usually has a heavy effect on consumers buying behavior in various product categories, including, games [58], movies [59], restaurants [60], and books [61]. Pelsmacker et al. [51] states that alongside reviews, also rating could impact consumers' attitudes. A number of studies take into account the rating as an element of reviews that also have some kind of effect on consumers [62,63]. Therefore, Ref. [59] demonstrates that consumers now have access to any information and exchange opinions on products, services, and companies easily.

As a result, more businesses begin to offer online services for consumers to write their reviews. Examples of such services are Amazon, Yelp, IMDb, Metacritic, and so on. Further, ABC, CBS, and NBC television networks provided a place for viewers to discuss their shows and programs online [59]. Hence, making the web more like a medium to reach people in order to generate more awareness of their products and services. Also, with the different types of reviewers' argument quality, readers could easily differentiate with certain reviews. Thus, many websites offer the ability for readers to rate reviews to be useful or not for them [64]. All these features made reviews more crucial every day. Especially at this time when all countries suffer from the pandemic and forcing people to stay in their homes, which made the dependence on online reviews higher than ever.

In summary, online reviews become more needed than ever due to different aspects, such as knowing the quality of the products, making the right decisions about what to buy. Moreover, feedback for companies, manufacturers, and owners, as well as staying at homes due to the COVID-19 pandemic situation (quarantine). These aspects or reasons increased the importance and popularity of online reviews. Therefore, the number of platforms that support online reviews increased. However, due to the widespread and simplicity of users who post and read reviews, it becomes more exposed to turn these reviews into negative opinions, namely, spam reviews. Also during the COVID-19 situation, people staring to concentrate on the pandemic on social media more than ever. Where most posts and tweets were about the feeling of the circumstances. Such behavior allows researchers to understand people's mental stress and illness, and then produces different analyses such as sentiment analysis. Thus, help people from panicking or deterioration of their mental state that could cause more increase in the number of bad decisions [65,66].

### 2.2. Spam Reviews

With the huge achievement of reviews recently due to its important and great features for many people. This success attracts a numerous number of individuals that their intention to spread false information and reviews for different objectives, including ruin or improve product reputation. This type of review is known as fake or spam review. Martens and Maalej [67] described spam reviews as a procedure done by fake reviewers that get paid for submitting such reviews. These reviews might or might not be actual users of the product. While, Ref. [68], stated that spam reviews as an act from writers, publishers, and vendors that post non-authentic online reviews to increase product sales. Further, Banerjee and Chua explained that spam reviews in tourism, for example, are similar to online reviews written by the imagination of several persons without the experience of going to that destination [69]. As for [44], they defined spam reviews as a review that is inconsistent with the genuine evaluations of services or products. Therefore, spam

reviews consider deceptive, false, and bogus reviews could be posted by various types of individuals, namely, review platforms online merchants, and consumers.

According to [70–73], the percentage of spam reviews ranges from 16%, 20%, and 25% to 33.3%, respectively. Many situations where the spam reviews had a role in causing damage to several parties. For instance, the UK Advertising Standards Authority reported that the TripAdvisor website was involved in generating more than 50 million spam reviews in 2012 [74]. While, in 2013, the Taiwan Federal Trade Commission ordered Samsung to pay a fine for spreading negative spam reviews [75]. Moreover, Amazon sued more than 1000 reviewers for posting spam reviews [76]. Mafengwo's website for a tourism platform in China was also involved in review deception [77].

Spam reviews have the ability to manipulate the market effectively. This is done due to being complex in structure and similar to real ones. However, spam reviews tend to be more influential. Various platforms try to manipulate and add spam reviews in order to boost traffic and deceive consumers into getting involved in the argument [78]. Therefore, spam reviews become more challenging to identify day after day. Spammers have been employed novel techniques in order to increase the difficulty to prevent and detect their spam reviews. Such techniques can be, for example, using a different style of writing periodically, duplicating real users reviews on other products, and implementing complex bots that can spread spam reviews.

Both academia and industry tried several countermeasures to mitigate the spam reviews phenomenon. Also, many governments imposed a number of laws and penalized anyone who performs such an act. In 2013, for example, the Attorney General of New York State led the operation "Clean Turf" for identifying companies that generate and post spam reviews [79]. In 2018, the Chinese government legislated the first E-commerce Law that prevents merchants from preform misleading and false promotions through posing spam reviews [80]. On the other hand, researchers proposed numerous attempts to detect spam reviews [81–84]. However, these attempts did not have much impact on decreasing the spread of all spam reviews. They offer different methods that can detect an exact type of spam review, notwithstanding, it's hard to identify all types. Hence, more approaches have been proposed periodically to handle new and specific types of spam reviews.

*2.3. Detection*

Detection can be described as a process to prevent and identify something that presents and further perform its purpose either being positive or negative [85–90]. Klein et al. [91] states that detection is a procedure that required action after the individuals in control become concerned about an event that might cause undesirable and unexpected activity. Whereas, Cowan explained the detection as an accumulation of several discrepancies that reached the threshold [92]. Furthermore, detection in the security fields is more critical than detection in other fields. For example, the authors in [93] described security threat detection as a practice of examining the security ecosystem in order to distinguish any malicious activities. The main principle of the detection process relies on the technology perception to discover unique patterns among all behaviors [94]. Besides, the intention of security detection is to implement automatic systems that are capable of investigating the possibility of different kinds of threats [95].

The detection process might help us with an early warning of the possible conditions that can occur. Thus, when we need a countermeasure for an exact incident, there will be time to construct a plan and execute the steps of that plan. For example, when a person involves in a routine activity like driving, he needs to observe the disturbances that could imply traffic jams or dangerous situations. Even in a stable circumstance (a tree branch could fall on a car or a maintenance procedure that might impact the safety of a plant operation). An alert of possible risk is desired in order to make the correct action to prevent the loss of important resources. As soon as the problem is detected, various procedures can be applied, including, collecting more information, monitoring the events closely, defining the problem, or discussing the situation with others.

Detection can be found in all fields, especially in security, such as malware, anomaly, intrusion, and spam detection. Malware detection, for instance, refers to the procedure of finding malicious software (e.g., viruses, spyware, and ransomware) on a system and trying to perform extensive damage or steal information of that system [96]. While intrusion detection is consists of a software application that attempt to monitor the system or network for any policy violations malicious activity [97]. In addition to that, anomaly detection is a process of recognizing the outlier or difference of normal activity, usually in a network. As for spam detection, it has more variants than the other detection methods. Spam can be in emails, social network profiles and posts, messages, and, reviews.

Spam reviews are characterized to be complicated due to their structure. Therefore, complex problems required complex solutions. In literature, many detection methods for spam reviews have been presented. [81] introduced a network-based spam detection model for online reviews in the social network environment, while [98] study the detection of spam reviews using the reviews processing and the rating of that reviews. Deep learning also has been used to identify the spam reviews through [99], and the authors of [100] investigated the spam detection for reviews by integrating the probabilistic review graph with the multi-modal embedded representation. Such methods are two years old and the style of spam reviews has been improved. Consequently, new and enhanced approaches become imperative. The discussion and analysis of the novel methods can be found in Section 3.

### 2.4. Datasets

A dataset in general is a collection of data that take the shape of tabular and can be found in every research domain, for example, security, medical, business, geoscience, etc. These datasets have various forms based on their structure and properties. The structure of the dataset is usually what determines the research problem, namely, supervised, semi-supervised and unsupervised learning. Technically speaking, without datasets machine learning can't perform and execute. Data is considered a significant part of machine learning models that required several procedures to be generated. Such procedures range from collecting, cleaning, pre-processing, formatting to labeling.

Regarding the spam reviews datasets, there are three types used commonly in the literature, content, meta-data, and product information. Each of which has its own advantage and disadvantage traits [15]. The content-type data, for instance, is consists of the review textual features. That is to say, it is a linguistic feature extracted from the review content. These features can be POS n-grams or words extracted for the purpose of identifying reviews' origin. Despite the fact that this type has a significant role in detecting spam reviews. The methods used are not sufficient to distinguish all kinds of spam reviews.

Further, the second type (meta-data) is more relevant for the review details besides its actual content, such as the identity of the reviewer, IP, and MAC, and IP addresses, rating, time of the review, geolocation place and review writing time duration. By examining such data, certain malicious behaviors can be identified. For example, when various user-ids post a number of negative and positive reviews of a certain product using the same device it shows suspicious behavior. Also, some reviewers write positive reviews for a particular brand and at the same time, negative reviews for other brands might consider doubtful reviews. Additionally, reviews for a specific hotel were found to be near its location; these reviews obviously not trustworthy because reviewers should be in other places. This type considers effective for spam reviews detection, however, the number of features is limited.

As for product information datasets, this type takes into account the product sales number and description to detect spam reviews. An example of such type, when a low sales product has too many positive reviews demonstrates the reliability of the reviews. Besides, there are some datasets that combined two or three types together in order to increase the performance of the detection. More experts can distinguish the spam reviews due to an increase in employed features.

Creating and generating spam review datasets consider very complex. Few researchers adopt the new alternative method for labeling and collecting artificial reviews. By creating

synthetic spam reviews datasets and taking existing real reviews to builds their model. An example of the context of the spam and real reviews available at [101].

Several numbers of the proposed spam reviews datasets in the literature are summarized in Table 1. Also, the distribution of the sources of the datasets can be found in Figure 2.

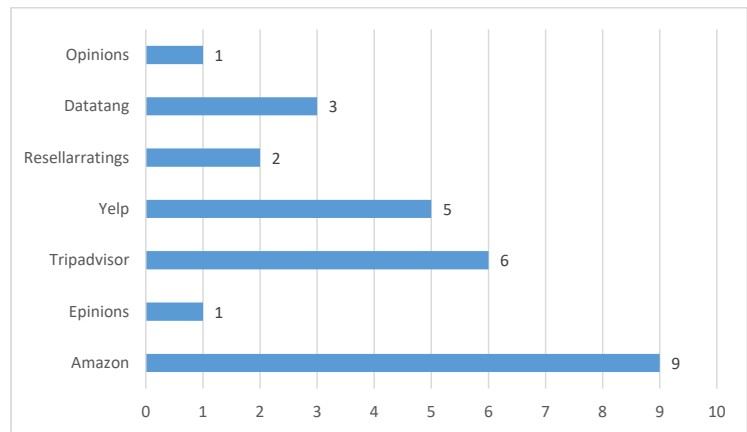

**Figure 2.** Number of datasets for each website.

**Table 1.** Descriptions of spam reviews datasets in literature.

| # | Source | Volume | Reference |
|---|--------|--------|-----------|
| 1 | Amazon | 5.8 million | [102] |
| 2 | Epinions | 6000 | [103] |
| 3 | Tripadvisor | 3032 | [104] |
| 4 | Yelp | - | [105] |
| 5 | Tripadvisor | 1600 | [106] |
| 6 | Tripadvisor | 2848 | [107] |
| 7 | Tripadvisor | 27,952 | [108] |
| 8 | Resellarratings | 628,707 | [109] |
| 9 | Resellarratings | 408,470 | [110] |
| 10 | Datatang | 10,020 | [111] |
| 11 | Datatang | 493,982 | [112] |
| 12 | Amazon | 65,098 | [113] |
| 13 | Amazon | 109,518 | [114] |
| 14 | Amazon | 195,174 | [115] |
| 15 | Amazon | 3 million | [116] |
| 16 | Amazon | 542,085 | [117] |
| 17 | Amazon | 6819 | [118] |
| 18 | Tripadvisor | 3000 | [119] |
| 19 | Yelp | 67,395 | [105] |
| 20 | Yelp | 359,052 | [120] |
| 21 | Yelp | 608,598 | [120] |
| 22 | Datatang | 9765 | [121] |
| 23 | Tripadvisor | 800 | [122] |
| 24 | Amazon | - | [123] |
| 25 | Amazon | 142.8 million | [124] |
| 26 | Yelp | 18,912 | [125] |
| 27 | Opinions | 6000 | [103] |

## 3. Approaches

Detection methods have emerged to identify and control the situation of spam reviews in industrial and academic institutions. One example of these methods is behavior detection based. This type of detection depends on users' behavior to differentiate them from spammers in order to detect their reviews. The work in [126], for instance, investigates this problem using Generative Adversarial Network (GAN). Where it generates synthetic features behavior of the new users. The work begins by choosing six real behavior features from normal users and training the GAN to generate synthetic features such as rating, attribute, and text features. Moreover, the authors implement a novel generator and discriminator for better training. The process helps the GAN to employ new users in order to generate the synthetic behavior features. The experiments were done on Yelp datasets and the results show that their framework outperforms the other methods. Another example of the behavior detection type is demonstrated by [127]. The paper presented a model to distinguish spam reviews from real ones. This model follows the procedure of an end-to-end training method to exploit the properties of random forest and Autoencoder. The implementation of the decision tree model is applied in order to identify the global parameter of the learning operation. Compared with other methods on the Amazon review dataset, this paper's experiments reveal that the proposed model achieves better results.

Furthermore, another detection method has also been presented in the literature, namely supervised learning-based. In this type, the labeled instances are used to learn using the machine learning technique. Several supervised learning-based approaches have been applied, one of them was the work in [128]. The authors investigated the detection of fake online reviews that can cause a severe impact on user decisions using supervised classification models. The contribution of this study is begun by extracting different features and then re-engineering them -using Cumulative Relative Frequency Distribution- to improve the detection phase. The evaluation process shows that their approach exceeds other methods on the Yelp dataset. On the other hand, Ref. [129] tried to resolve the falsification of online reviews by using supervised classifiers techniques. This attempt to handle such a problem employed neural approaches to classify the reviews by training the models on syntactic and lexical patterns. The work proceeds by comparing various types of supervised classification models. They applied Google's latter architecture (BERT) in order to detect fake reviews. The approach achieves 90% in terms of accuracy. Likewise, Ref. [130] introduced a detection method (HOTFRED) to identify the online fake reviews in the Tourism Domain. Hoteliers and guests suffer from these kinds of reviews when trying to plan or select the optimal hotels for their journey. HOTFRED considers a dynamic hotel detection system for fake reviews using different analytical approaches. In spite of its excellent ability to detect fake reviews, the system can also serve as an automatic tool for hoteliers to ensure the hotel they choose is the right one for their requirements.

These aforementioned approaches for both behavior and supervised detection based used widely for detecting spam reviews in literature. Further, many other approaches have been also applied for reviews detection by different researchers, which are addressed in detail in the following subsections. These approaches are more specific and described based on their category, namely Semi-supervised, Deep Learning, Linguistics, Ensemble, Graph-based, Sentiment, and Other approaches.

### 3.1. Semi-Supervised

In this first subsection, the spam online reviews based on the semi-supervised method are discussed. A semi-supervised approach or learning is a technique that operates on a dataset that combines labeled and unlabeled instances [131]. That is to say, the semi-supervised method is a merge between unsupervised learning (no labeled instances) and supervised learning (labeled instances). Usually, the amount of unlabeled parts of sets exceeds the number of labeled sets by far.

In literature, this method obtained attention between researchers with its performance to handle different types of datasets. For example, Ref. [132] argues that there are many works presented to solve the identification between real and spam reviews. However, the issue is still challenging to manage for supervised learning due to the lack of labeled data samples as well as imbalanced problems. Therefore, the authors see that using a semi-supervised learning approach is better for spam reviews problem. Their approach-Ramp One-Class SVM- consists of a nonconvex semi-supervised method. Which operates on one-class data in order to deal with the lack of labeled datasets. Also, solving the outliers and non-review using the nonconvex properties of the proposed approach loss function. The experiments were executed on two datasets, namely Yelp and Ott. The outcomes show that the proposed method outperforms other techniques in terms of accuracy, precision, and recall.

Moreover, the work in [133] also studied the performance of the semi-supervised method on spam reviews detection. The hybrid semi-supervised learning approach that they presented relies on the user–product and users' characteristics relations. In their work, the hybrid PU-learning-based spammer detection (hPSD) starts its detection process by producing various positive samples. Then the semi-supervised learning classifiers are evaluated on the movie dataset. The hPSD achieves the best results when compared with other approaches. Furthermore, the approach used the real-life Amazon spam reviews detection dataset for more examinations. Another recent framework proposed by [134] for spam reviews detection using semi-supervised learning. The work simply used the adversarial training mechanism that exploits the Generative Pre-Training 2 (GPT-2) abilities to detect spam reviews against unlabeled and labeled data. The experiments were performed on the TripAdvisor and YelpZip datasets. The presented model obtained the highest results with 7% more in terms of accuracy. Moreover, due to the lack of labeled data, the model generates synthetic samples (spam/non-spam reviews) to prove more labeled instances to learn better.

Ligthart et al. [135] stated that in real-world situations the datasets that are used usually lack the required labels to perform on supervised models. Therefore, applying the semi-supervised learning approaches can be more efficient. The purpose of this study is to explore the effectiveness of the various type of semi-supervised learning as a classification method. The authors examined four different semi-supervised models in order to classify the spam online reviews for hotels. Additionally, the experiments comparison shows that the Naive Bayes acquire the best results with 93% accuracy. The study demonstrates the need to mitigate the labeling efforts while focusing more on model performance when the labels are limited. Whereas, Ref. [136] introduced a dictionary based on online reviews and social network terms to find the hidden pattern and relationship of these terms. A number of language features were used, including, length of reviews, presence, and frequency of bigram, and presence and frequency of unigram. The work employed both supervised and semi-supervised methods to detect spam reviews with the help of linguistic and behavioral features.

According to previously addressed studies, the semi-supervised approach is desired the most when the data are not fully labeled. Such scenarios can usually occur in real-world situations. Therefore, these kinds of approaches are important to overcome the problem of limited labeled data.

### 3.2. Deep Learning

The idea of the deep learning concept came from the way the human brain function to process data and generate patterns for decision making. Deep learning is known to be a subset of machine learning that is capable of preform classification tasks on different data types, namely text, images, videos, and sounds. Recently, deep learning techniques gain attention in different applications, especially spam reviews detection.

Due to its ability to learn from big data, many authors proposed various approaches based on deep learning to solve spam review detection. One example for this is the work in [137], where a Convolutional Neural Network (CNN) is presented. CNN is applied in

order to identify the semantic information of users reviews that can enhance the detection of deceptive characteristics. When compared with other neural network architectures' performance. The proposed CNN operates better in detecting spam reviews. Similar to the prior work, the authors of [138] also argue on the superiority of deep learning techniques in extracting the semantic aspect of reviews context. Their new approach (Paragraph Vector Distributed Bag of Words (PV-DBOW)) can recognize the global representation of reviews semantics. Furthermore, the representations of reviews transfer to a neural network approach to detect spam reviews. The experiments show that the PV-DBOW outperforms the existing state-of-the-art methods.

On the other hand, Ref. [139] introduced a novel multi-dimensional features approach for spam reviews detection. The approach used the standard component in order to obtain the low-dimensional features to classify the user-product connection. Long short-term memory (LSTM) is applied and trained with a capsule network to identify the spatial structure and textual context features. Moreover, the model merges the user behavioral and text features to be utilized as input for the detection classification module. The outcome of the approach proves its ability to detect spam reviews more than the existing methods. Besides, the study of [140] debates the lack of labeled dataset availability in spam reviews. Consequently, the authors suggested LSTM networks based on unsupervised learning to discriminate the spam reviews from the real ones. The model is trained to discover the patterns of real reviews without the required labels. The experimental results reveal that their framework can distinguish spam from real reviews efficiently. Zhou and Zhang [141] also employed the LSTM to study the semantic features of spam reviews. The authors used Deep Belief Network to detect the credibility of product reviews and CNN for discrete features extraction. Thus, connecting the traditional features with semantic features to build a DBN model. The performance of deep confidence network model performs better compared with the standard machine learning methods.

Likewise, the work in [142] expressed its thoughts about how the deficiency of traditional machine learning methods to detect spam reviews given limited feature representations. As a result, a Deep Learning (DL) framework is presented to detect spam reviews. The DL framework combined with Self Attention-based CNN BiLSTM (ACB) to extract and identify the document representation of such reviews. Further, the weights of each word are calculated and learning the spamming clues in the sentence and document, respectively. Afterwords. the model studies the sentence representation through CNN as well as discovers the n-gram features of a higher level. In the end, the vectors of sentences are combined by using Bi-directional LSTM in order to spam reviews detection based on contextual information. The experimental results show that the ACB achieved the highest results compared with other variants methods in terms of accuracy.

The works that were previously stated in this subsection indicates and demonstrate the powerful performance of deep learning techniques to identify the semantic structures alongside the hidden pattern of spam reviews. In addition, the capability to generate specific features for real and spam reviews based on the textual context. Such works consider essential, particularly when analyzing the reviews' behavior based on their context traits and features space.

### 3.3. Linguistics

Linguistics or multilingual approaches usually rely on the contextual structure of the text of reviews. The lingual detection systems prove that the issue also occurs in other regions and languages. Furthermore, this kind of system depends more on the language characteristics itself, so most features are text-based (linguistic features).

Hussain et al. [143] declared that most of the spam review detection studies are more towards languages such as Chinese, Arabic, and English. Hence, in their work, they aim to develop a spam review detection approach based on the Roman Urdu language. The approach is implemented on several classification models based on two types of features, including behavioral features and linguistic features. Three different perspectives were taken into account when the performance was evaluated. First, the linguistic features

only were utilized in the classification models. Second, the behavioral features merge with non-distributional and distributional facets for evaluation matter. While the third perspective, both linguistic and behavioral features are combined together and employed for evaluation. The experimental evaluations showed that the best approach was the one with both features combined (linguistic and behavioral) with 0.96 in terms of accuracy. Further, another crucial objective is considered. Increases the trust of the reviews and mitigates the spam reviews from spreading in the South Asian region. Additional recent work also applied the spam review detection method based on the behavioral and linguistic features [144]. Similar to the previous study, two different methods were used to detect spam reviews. First, by utilizing the Behavioral Method (SRD-BM), while the second by utilizing the Linguistic Method (SRD-LM). The SRD-BM used 13 different features, whereas RD-LM considered the textual features for spam reviews detection. The results demonstrated the performance of both methods compared to other state-of-art approaches with 93.1% and 88.5% for SRD-BM and SRD-LM, respectively.

The authors of [145] investigated various supervised machine learning to detect spam reviews based on linguistic content-based and Word Count (LIWC). In order to reduce the huge dimensional of the data. Principal Component Analysis (PCA) technique is used. Five different variances with and without PCA variances were employed for the evaluation process as well as several machine learning techniques were applied. The Ensemble Bagged classifier outperforms the other supervised methods with 88% in terms of accuracy. Moreover, the work in [146] presented an unsupervised approach for spam reviews detection on videos, images, and Chinese texts. Their approach outcome various findings after the evaluation process: 1. The amount of image spam is more than the video and text spam; 2. Stealing from reviews is more appealing than just borrowing something from the marketing; 3. Spammers use fictitious rare incidents more than any tricks type in order to influence customers; 4. Utilizing the same methods for texts images and videos is common.

Ansari and Gupta [147] reported that the way how customers understand spam reviews is still not investigated enough. Hence, they designed a theoretical approach that describes the linguistic style of the reviewer's intentions. This approach was evaluated on 120 reviews that were examined by applying the fractional logit model. The conclusions of the results demonstrated that the method used by the speaker shows his intention. Also, reviews with lower argument structuring, flattering and contextual embedding are comprehended by customers as spam reviews more than other reviews. Furthermore, Ref. [148] presented two study for spam reviews detection. In the first study, they used the Yelp dataset that has Linguistic Inquiry Word Count (LIWC) features. While the second study, the reviews were evaluated using 660 participants to label reviews as confident or doubtful. The outcome of both studies suggested that positive reviews were less doubtful than negative ones. This study highlights the advantages of showing doubt. In addition, the study in [149] develops a machine-learning model to identify opinion trustworthiness. The authors generated a large-scale dataset of spam reviews, with 869 deceptive and 866 truthful reviews. The dataset was in the Korean language which was the first attempt for reviews in such language. The results reveal that the model achieves about 81% in terms of accuracy.

In this subsection, the works attempt to solve spam reviews detection based on multilingual cases. These detection linguistic systems are focused to resolve spam reviews in a specific region. Therefore, they are considered important to handle problems with different languages, regions, and cultures.

### 3.4. Ensemble

Ensemble methods are known as machine learning techniques that consist of a set of classification models as shown in Figure 3. The point of aggregating them together is to take into account their weighted predictions as a vote. This type of machine learning method is performed in order to enhance the classification accuracy and depend more on several models rather than one model (best model). Fayaz et al. [150] presented an ensemble approach that combines different classifiers, which are, Random Forest (RF), Multilayer Perceptron (MLP), and K-Nearest Neighbour (k-NN) for detecting spam reviews. Their



ensemble model was evaluated on the Yelp dataset alongside using different types of feature selection methods to select the best subset of features. The proposed approach outperforms the individual classifiers (RF, MLP, and k-NN) and state-of-the-art methods.

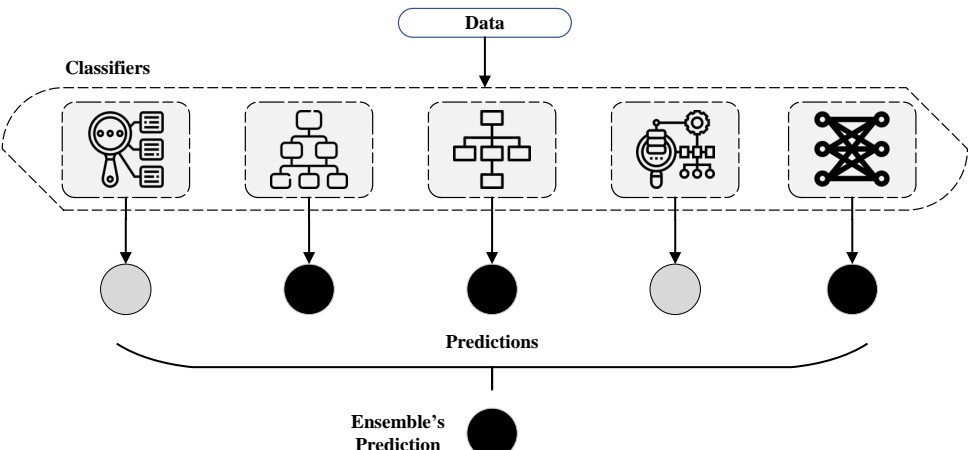

**Figure 3.** Ensemble learning architecture.

Moreover, Ref. [151] proposed a novel study for spam reviews detection using ensemble machine learning methods. In order to improve the accuracy of classification, an additional study was performed to analyze features in combination with ensemble methods. Further, four ensembles are used, including, Extra Tree, Bagging, Random Forest, and Boosting. The ensemble method obtained the best results compared with other methods. Another recent work applied the ensemble model for the identification of spam reviews [152]. Four different steps are used in this study, scilicet, Data resampling, Feature pruning, Parameters optimization, and classifier ensembling. Then the resampling technique is employed to solve the imbalance problem. The results verified the performance dominance of the model against other methods in spam reviews detection.

Additionally, the authors of [153] introduced a spam reviews detection method based on the ensemble machine learning technique. Three various models are given and trained on multi-view learning techniques to ensemble them together for majority prediction. Then they extract the text information of the reviews through parallel convolution neural networks (CNN) and bag-of-n-grams. The CNN architecture employed the n-gram embeddings as input and extract feature representations of reviews by using parallel convolutional blocks. The evaluation phase takes place on Yelp Filtered dataset with a 92% rate in terms of F1 scores. While the study of [154] built a detection framework for spam reviews using single and ensemble models. Also, the work proposed two random sampling methods for solving the class imbalance issue. The experimental results show that the sampling methods (random under and over-sampling) enhanced the classification accuracy of spam reviews. Besides, the Adaptive Boosting ensemble obtained the highest results for smaller datasets, while single classifiers achieved best when the dataset was larger.

The indication of the previous works suggested that using more than one classification model achieved better results in detecting spam reviews. In other words, taking the vote weighting of several classification models is better than relying on the best model, where the performance is more reliable due to the fact that various decisions are taken.

### 3.5. Graph-Based

Graph-based spam reviews detection is a process of mathematically representing the network and their relationship through lines and points to describe the connection of the reviews entities. That is to say, the graph-based approach is a branch of data mining techniques that depend on the inter-dependencies between objects. This approach is used to study the patterns of the relations of a network in order to distinguish the hidden patterns. In our case, graph data of the spam reviews (fraudulent) rely on the rating of the reviewer

for a product and the other reviewers' rate for that product. While the non-spam reviews (trustworthy) rely, somehow, on different products that they rated. Therefore, graph-based detection introduced a robust mechanism for capturing and identifying the relationships and correlations of data objects inter-dependent.

For instance, Ref. [155] provided a novel graph-based spam detection model to recognize the reliability of reviews. The paper compared various methods for the sake of proving the efficiency of the proposed model. Different attack scenarios are taken into account when examining the model. The results indicate that the proposed model identifies the spam reviews in different scenarios efficiently. Moreover, it can reduce the trust value of the spam review as well as prevent spammers from ruining the reputation of the product. Sundar et al. [156] used a deep dynamic clustering model to detect spam reviews by utilizing graph embedding structure to preserve the nonlinear information structure of the text. The authors also applied dynamic aspects of reviewers in order to determine the spam users from the normal ones. In the phase of the experiment, the results show that the spam detection model reached 92%. While [157], stated that the current spam reviews detection techniques used one to two kinds of entities only alongside employing a few types of features, namely content, relation, and behavior features. However, such techniques suffer from not being able to be employed in context and tend more to synthetic criteria. Therefore, a new graph-based model is presented to detect spam reviews. The model (Multi-iterative Graph-based opinion Spam Detection (MGSD) is utilized with all classes of entities as well as a unified structure. Afterward, both the implicit and explicit relation is used, then an evaluation of the Spamicity effects is applied. In order to improve the performance of the detection model number of weighted features were taken into account. The outcome of the experiments indicated the superior of the proposed model with 93% and 95.3% for the synthetic dataset and Ott's dataset, respectively.

Further, Ref. [158] declared that the previous works in the literature consider several reviews as a whole when performing feature extraction, at the same time, disregarding the internal differences and similarities. Thus, causes to not identify the most discriminative information due to semantics reviews disorder. To solve such a problem, they proposed a co-attention framework and orthogonal model for the decomposition process. Hence, learn the differences and similarities between the reviews. Also, to distinguish the strong social connection, they determined first the weak relation graph through the dynamic interactions method. Subsequently, graph representation is applied to learn the interaction of social connections. Their framework outperforms the state-of-the-art methods evaluated on two real-world datasets. Whereas, the authors of [159] designed a community detection technique (SC-Com) for spam reviews detection. SC-Com used the graph of reviewers for splitting the communities based on their reciprocal doubt. Then a temporal abnormality and community-based features extraction are employed for identifying the spam reviews from non-spam. Using a real-world dataset, an evaluation phase is done and the presented approach achieves the highest results utilizing ratings and time data.

The graph-based approach relies on the relationship between different entities of the review process, namely review, reviewer, rating, and so on. Hence, non-labeled data is needed and low complexity of the structure review compared to other approaches. This type of detection method facilitates the obtain of relation and interaction hidden-pattern between the network. Accordingly, making graph-based detection techniques different from other methods. As a result, it depends on other factors which aren't accessible in other cases.

### 3.6. Sentiment-Based

Sentiment analysis (SA) can be defined as a natural language processing method employed in most cases for the sake of knowing and understanding the content of text emotion, including negative, positive, and neutral. SA is usually used as feedback for business products in order to improve the quality, defects, and price of a product as well as understand customer needs.

A set of recent works applied SA as a mechanism to detect spam reviews. For example, Ref. [160] designed an opinion spam detection framework based on SA. The framework adopted the Persian language to identify the spam reviews and generate novel features related to that language characteristics. Using AdaBoost and Decision Tree the classification accuracy reach 98% and 98.6%, respectively. Besides, the newly created features were utilized and compared with other features set from literature. Patil et al. [161] stated that opinion mining is generally performed for SA recognition. However, not all data can be sure of its trustworthiness. As such, it's important to detect spam reviews. Therefore, this work presented a product-based SA and a spam identification model for online reviews (ALOSI). Consequently, emphasize the dissimilarity of opinion sentiment with and without considering spam reviews.

Furthermore, the authors of [162] developed an oriented sentiment mining approach able to recognize spam reviews based on designated topics. The findings show that their method exceeds different state-of-the-art techniques based on two aspects burstiness of time and content duplication. While, the study introduced by [163] examined several facets, which are language, rating sentiment, and content. Also, an extra investigation has been provided to explore the deception and behavior for detecting spam reviews. Above 20 features have been extracted to characterize reviews and multiple machine learning models evaluated on spam reviews detection. The outcome of this study suggests that there are discrepancies in reviews and have plosive impacts on the spam detection performance. Another recent framework devolved for the sake of detecting spam reviews using the opinion mining method [164]. Two machine learning models have been proposed, one for spam reviews detection and the other for knowing the rates of these reviews. The models were designed using random forest and Naive Bayes techniques. Yelp dataset was considered in order to evaluate both models for the purpose of spam reviews detection and opinion mining.

In certain scenarios of sentiment analysis applications, knowing reviews' reliability is considered crucial for different parities. Various methods have been employed to combine the two methods as seen above. Such studies achieved another level or additional aspect of just knowing the sentiment of the reviews but also comprehending if ts spam or not. As a result, more understanding of the type of reviews and their properties in various cases. Nevertheless, more study needs the detection of spam reviews using SA techniques.

### 3.7. Other Approaches

This section addresses various works that have a general theme and don't belong to one of the aforementioned categories. Most of them focus on solving spam reviews using different detection systems, such as Content-Based detection, concept drift benchmark-detection, epistemic belief, and unsupervised learning.

For instance, Ref. [165] implemented their detection system through a collective technique to identify spammers and spam reviews (adversarial type). The process began by training two models Content-Based Module (CBM) and Behavior-Based Module (BBM). Afterward, both models were co-trained in order to receive each other feedback. According to their results, their method performs better in determining the adversarial reviews. Another example of this category for detecting spam reviews is proposed by [166]. The work developed a novel automated tool for identifying spam reviews using 1041 respondents. Then a comparison of the non and spam reviews with psycholinguistic deception cues. The findings show that the tool obtained an 81% in terms of accuracy.

Moreover, Ref. [167] presented a topic model and reviewer anomaly rate method for spam reviews detection. The authors split the spam reviews into deceptive and content-type. In the first part, the dataset is modeled through an LDA topic that observes the reviews' content type, while in the second part, the abnormality degree index is used to recognize the deceptive reviews. A score was assigned for each review and then merged with adaptive weight calculation according to similarity and abnormal of the reviews. High score reviews consider spam, whilst low score review is recognized as non-spam. Mohawesh et al. [168] investigated the drift problem in spam reviews using two techniques,

namely content-based classification, and concept drift benchmark-detection techniques. Four real-world datasets were employed for the evaluation matter, the results show that there is a negative correlation between the performance of spam detection and concept drift.

The authors of [169] proposed a study about online reviewers' authenticity detection using algorithmic-based. The study takes into account the epistemic belief role, which is the person's ability to decide between facts and falsehood. More than 300 participants were getting involved in order to classify reviews, spam, and authentic. Using epistemic belief and some kind of justification to reduce the relationships between specificity and exaggeration. While the work in [170] declared that feature selection is one of the important methods to enhance the classification of spam detection and reduce the computation time of the training phase. Hence, they study the impact of various types of feature selection methods in detecting spam reviews. Several feature selection methods have been applied and trained on four classification models. Additionally, an examination performs on three well-known datasets alongside the different types of feature selection methods, including, bigram, unigram, word embedding, and frequency count. Their experimental results prove the effects of using different factors on classification performance. Furthermore, Ref. [171] proposed a detection approach for online reviews manipulation. The authors collected a number of reviews from 500 doctors, consisting of textual feedback and ratings (1 to 5). Their study explored the procedure to verify the negative reviews, then rank the doctors and reduce risks in the healthcare environment.

Additional recent research about pseudo-reviews detection presented by [172]. Two studies have been done for the purpose of identifying such reviews. The first study concentrates on ensuring the slight impact on product attitude when it's in an isolation environment. In the second study, both pseudo and authentic reviews combined together, where shows that the impact was negatively on purchase intentions. On the other hand, the work of [173] discussed the aspects of how to increase the confidence in reviews. Also, the study analyses two review sites and debates their trustworthiness, namely TripAdvisor and Booking. Three types of analysis were provided, verification analysis, SWOT analysis, and processual analysis. Their model provides verification of tourism services reviews in destination and reviews.

Besides, few works applied the unsupervised learning technique for spam reviews detection. One example for this was proposed by [174], where the paper consecrate on an abandoned domain (movie reviews) and implemented a new unsupervised spam identification model based on attention mechanism. Also, a statistical features extraction was performed as well as introducing an attention mechanism for review embedding and applying conditional generative adversarial network for training the model. The results demonstrate the outperforms of the model compared with other state-of-art approaches. Another example that employed the unsupervised learning technique suggested by [175]. The authors stated that the unsupervised machine learning methods can use the clusters as features. This could help the classification model's performance during the features reduction. In other words, the classification models improved when it executes after the clustering process. The outcomes of the experiments prove the enhancement on the SVM classifier when K-means clustering was applied earlier. Further, different feature selection methods are utilized to seek the optimum performance from the classification model. The publishers and type of studies of the reviewed works are summarized in Figures 4 and 5.

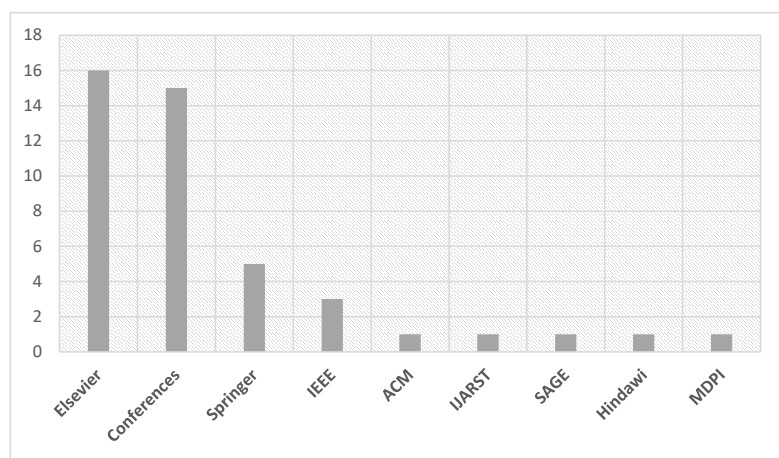

**Figure 4.** Number of papers reviewed in this study for each publisher.

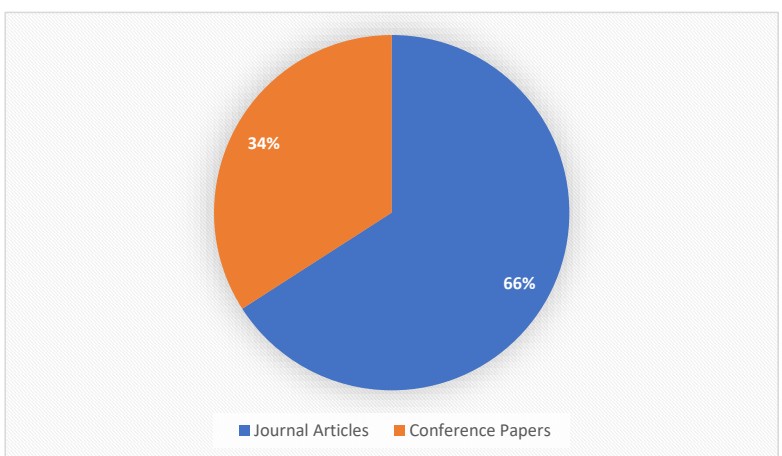

**Figure 5.** Type of selected studies that described in this work.

Two bibliometric analyses have been performed to provide a quantitative analysis of the reviewed article. Figures 6 and 7 summarize the metadata extracted from the papers' database. Figure 6, as can be seen, shows the number of published papers for each country. The numbers demonstrate the superiority of China, India, and the USA, where the least countries were France, Turkey, Saudi Arabia, Indonesia, Czech Sri Lanka, Iraq, and Malaysia with one article each. As for the number of citations, the first countries were China, France, and India with 101, 44, and 41 citations, respectively as can be noticed in that same figure.

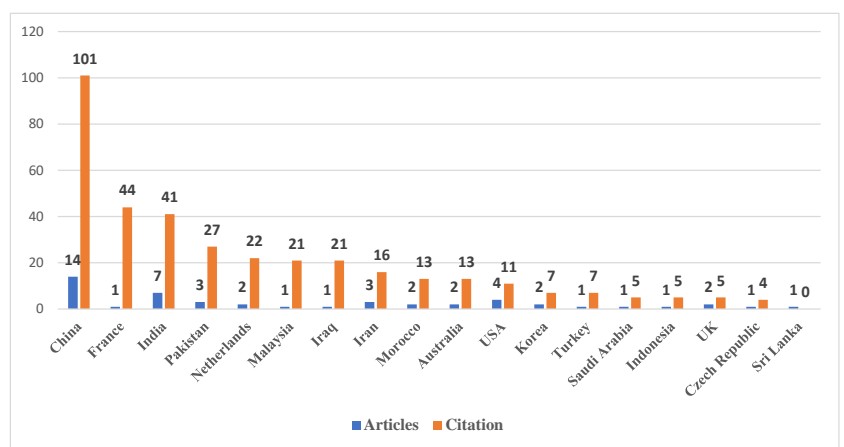

**Figure 6.** The number of articles and citations for each country of the reviewed studies.

Additionally, the most frequently occurring keywords for these articles are shown in Figure 7. The most prominent keyword was 'machine learning' (10 times), followed by 'Opinion mining' (8 times) and 'Opinion spam' (7 times). In terms of least appearance, the keywords were Social Behavior, Spammer detection, and Sentiment analysis.

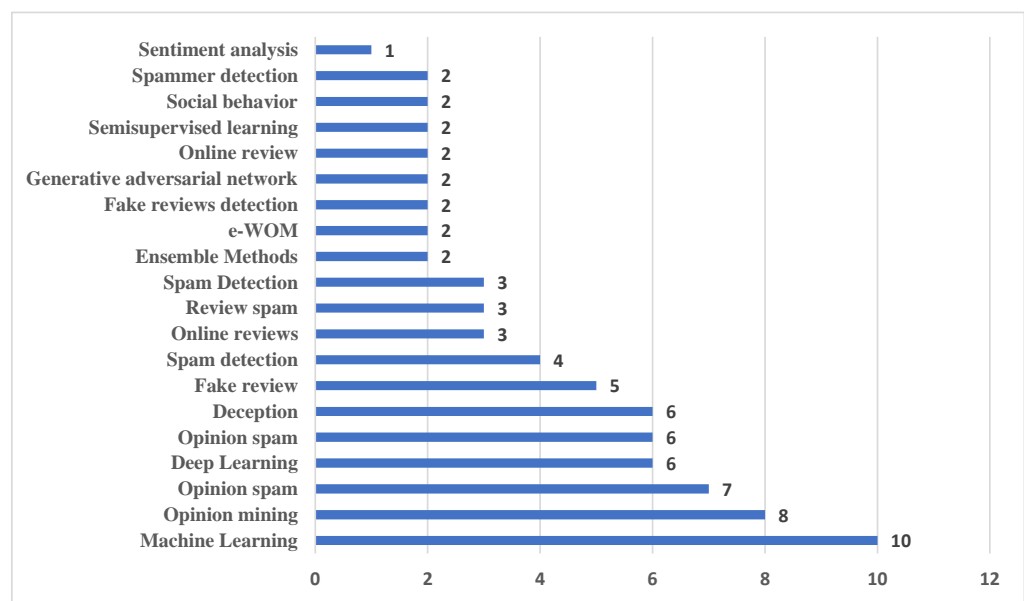

**Figure 7.** The most keywords appearance form the reviewed studies.

In this last subsection, we addressed the various type of work in order to mitigate and prevent spam reviews detection. These works range from using behavior, supervised and semi-supervised learning, deep learning, linguistic, ensemble, graph-based, sentiment-based, and theoretical studies that don't belong to any previously referred categories. Also, we applied two different bibliometric analysis for the reviewed papers.

## 4. Literature Analysis, Discussion and Future Directions

In this section, detailed analyses and interpretations of the presented works are proposed. These studies have their advantages and disadvantages. Therefore in this discussion, we will investigate their definition, strengths style, and weakness.

The first genre of the studies focuses on works that depend on behavior features. These type of features considers important and essential when trying to identify spam reviews. Due to the fact that they rely on the attitude of the spammers themselves. As such, many researchers combined this method with their main approach thanks to its ability to detect some kind of reviews. Examples of such behaviors could be the style of writing, the time of posting, rating of each review, name of their users, and so on. However, the behavior features technique could be tricked by using the correct methods (for example, changing the style of writing) without considering other combined schemes with this technique, such as machine learning methods.

As for the second genre, the supervised learning detection model is a well-known technique applied not only for spam reviews but also in different security and other domains. Supervised learning has a powerful ability to recognize the hidden pattern of reviews, including the relation of reviews, reviewers, products, features, and labels between each other. Such a pattern can help improve the identification of spam reviews efficiently. Hence, it is considered one of the best methods to detect spam reviews recently in the literature. Nevertheless, this technique has an essential weakness when performing. The lacking of many labeled data and the preparation of the data. This is happened due to the fact of labeling and pre-processing of the data consume time, effort, and money.

Further, the semi-supervised learning method attempted to solve what supervised learning lacked. Handling unlabeled data can be resolved to utilize the traits of semi-supervised learning without any problem. Semi-supervised can perform with a few numbers of labeled instances while the rest of the instances are unlabeled. That's why semi-supervised learning consider relatively inexpensive, where no need to consume money and time on labeling the whole dataset. In other words, it has the traits of both the supervised and unsupervised learning methods. On the other hand, the semi-supervised technique has two main disadvantages, which are the instability of iteration results and usually obtaining low accuracy.

Deep Learning is considered the future of machine learning, where it can detect spam reviews efficiently and dynamically. Particularly in spam reviews, where the construction of features is a necessity. Deep learning can handle different types of features, applications, and datasets. Thus, can easily outperform other approaches in spam reviews detection. However, deep learning requires huge data (big data) to perform and outperforms other methods. Also, it consumes a lot of time to train the model. Therefore, it needs more effort to deal with when optimal performance is the objective.

Moreover, linguistics approaches usually depend on the language characteristics and the context used in the reviews. Also, it addresses other languages than English alongside various regions to detect spam reviews. These languages and regions have different characteristics. Hence, it is important to have such a detection approach that can handle the linguistics, context, culture, regions, and multilingual spam reviews. But this method is hard to perform without the proper persons that can understand the used language. Also, it lacks the diversity of linguistic spam reviews detection methods.

The ensemble machine learning-based achieved excellent performance for spam reviews detection. Either using the voting method or the stacking method (a new classifier learn from the previous one). The ensemble can handle complex and difficult problems that need multiple hypotheses as well as it is unlikely to have an overfitting issue. On the negative side, the ensemble models are hard to interpret and understand the predictions (why these reviews are spam) due to their complexity. Such complexity is also computationally expensive and needs time and memory to perform.

While the graph-based spam reviews detection method consecrates on the relationships of the entities. In this scenario, these entities consist of reviews components and their relationship with each other. Thus, the detection phase of the graph-based depends on a low structure complexity alongside non-labeled data required. This method also has a number of advantages, such as summarizing huge datasets visually, the ability to compare several datasets, and can identify trends accurately. However, the graph-based method needs more analyses to perform than other approaches, where data misinterpretation could occur easily (ignoring important info, overlooking some prior knowledge, and focusing on irrelevant data) without taking the proper procedures.

Finally, the spam reviews detection based on sentiment analysis utilized the natural language processing technique to operate. This kind of detection attempts to understand and interpret the textual content in order to assist the process of spam reviews identification. Therefore, knowing different and unique aspects can improve the interpretation of some reviews content that is hard to understand on other methods. Nevertheless, combing sentiment analysis for spam reviews detection is not mature enough and needs more investigation. Hence, some wrong perceptions could take place. Table 2 illustrates the advantages and disadvantages of the aforementioned approaches.

In summary, there are many ways to detect spam reviews. Some of them are mature while others require more examination and study. Each approach (category) has its own advantage and disadvantage. Hence, it is good to use different approaches depending on the problem we face. Some methods can handle small data and others can handle the large ones, several requires labeled data and others don't, a few of them need more time to perform while others are needless, and so on. Thus, as mentioned previously the problem is what helps us to select the proper approach, not the other way around.

**Table 2.** Advantages and disadvantages of the reviewed approaches.

| Approach | Advantages | Disadvantages |
|---|---|---|
| Behavior features | Identify spam reviews by using the attitude of the spammers. | Modifying users' behavior (changing the style of writing). |
| Supervised learning | Powerful to recognize the hidden pattern of reviews, including the relation of reviews, reviewers, products, features, and labels between each other. | Lack of many labeled data and the preparation of the data. |
| Semi-supervised | Solve the supervised learning weakness, handling unlabeled data. | Instability of iteration results and obtain low accuracy. |
| Deep learning | Handling different types of features, applications, and datasets. | Requires huge data (big data) to perform. |
| Linguistics approaches | Handles different linguistics, context, culture, regions, and multilingual spam reviews. | Hard to perform without the right persons that can 726 understand the used language. |
| Ensemble | Can handle complex and difficult problems that needs multiple hypotheses. | Hard to interpret and understand the predictions (why these reviews are spam). |
| Graph-based | Consecrate on the relationships of the entities. Depends on a low structure complexity alongside non-labeled data required. | Needs more analyses to perform than other approaches (data misinterpretation could occur easily) |
| Sentiment-based | Knowing different and unique aspects can improve the interpretation of some reviews content that is hard to understand on other methods | Not mature enough and needs more investigation |

We also noticed that there are differences between the spam reviews studies before and after the COVID-19 situation. The rise of people staying indoors is responsible for the increase of reviews is an example. Between the two periods, different behavior of reviews is evident when using products and services or customers' target, context, and impact. With more people reading and searching for reviews, spam reviews are also growing in number and, models of detection techniques improve and become more diverse.

Table 3 summarize the details of the prior addressed works, based on several elements. Also, the word cloud of keywords used in spam reviews is illustrated in Figure 8.

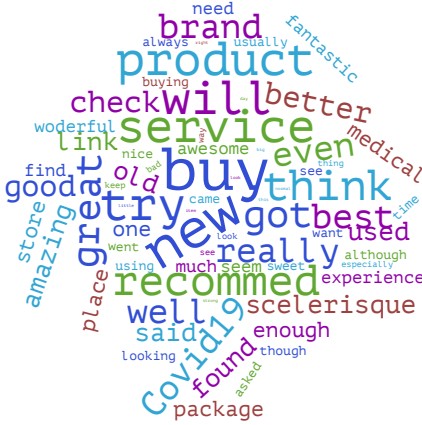

**Figure 8.** Word Cloud for spam reviews in different datasets.

**Table 3.** Detailed information of spam reviews detection works.

| # | Method | Category | Publication Channel | Year | Reference |
|---|--------|----------|---------------------|------|-----------|
| 1 | GAN | Behaviour | Journal (Elsevier) | 2020 | [126] |
| 2 | DT | Behaviour | Journal (Elsevier) | 2020 | [127] |
| 3 | CRFD | Supervised | Journal (ACM) | 2021 | [128] |
| 4 | NN | Supervised | Conference | 2020 | [129] |
| 5 | HOTFRED | Supervised | Conference | 2021 | [130] |
| 6 | Ramp One-Class SVM | Semi-supervised | Journal (Elsevier) | 2020 | [132] |
| 7 | hPSD | Semi-supervised | Journal (IEEE) | 2020 | [133] |
| 8 | GPT-2 | Semi-supervised | Journal | 2021 | [134] |
| 9 | Self-training | Semi-supervised | Journal (Elsevier) | 2020 | [135] |
| 10 | Hadoop | Semi-supervised | Journal (IJARST) | 2020 | [136] |
| 11 | CNN | Deep Learning | Conference | 2020 | [137] |
| 12 | PV-DBOW | Deep Learning | Journal (Elsevier) | 2020 | [138] |
| 13 | LSTM | Deep Learning | Conference | 2020 | [139] |
| 14 | LSTM | Deep Learning | Journal (Springer) | 2020 | [140] |
| 15 | LSTM + DBN | Deep Learning | Journal (Springer) | 2021 | [141] |
| 16 | CNN-BiLSTM | Deep Learning | Journal (Springer) | 2021 | [142] |
| 17 | Soft Voting | Linguistics | Journal | 2020 | [143] |
| 18 | SRD-LM | Linguistics | Journal (IEEE) | 2020 | [144] |
| 19 | PCA | Linguistics | Conference | 2021 | [145] |
| 20 | Duplication | Linguistics | Journal (Elsevier) | 2021 | [146] |
| 21 | Speech Act Theory | Linguistics | Journal (Elsevier) | 2021 | [147] |
| 22 | LIWC | Linguistics | Journal (Elsevier) | 2021 | [148] |
| 23 | SVM + DNN | Linguistics | Journal (SAGE) | 2021 | [149] |
| 24 | RF, MLP, and K-NN | Ensemble | Journal (Hindawi) | 2020 | [150] |
| 25 | ET, RF, Bagging and Boosting | Ensemble | Journal | 2020 | [151] |
| 26 | Lightgbm,RF and GBDT | Ensemble | Journal (IEEE) | 2021 | [152] |
| 27 | CNN | Ensemble | Journal (Springer) | 2021 | [153] |
| 28 | Adaptive Boosting | Ensemble | Journal (Springer) | 2021 | [154] |
| 29 | ROSD | Graph-based | Journal | 2020 | [155] |
| 30 | DDC | Graph-based | Conference | 2020 | [156] |
| 31 | MGSD | Graph-based | Journal (Elsevier) | 2020 | [157] |
| 32 | WS | Graph-based | Conference | 2020 | [158] |
| 33 | SC-Com | Graph-based | Journal (Elsevier) | 2021 | [159] |
| 34 | DT | Sentiment-based | Conference | 2020 | [160] |
| 35 | ALOSI | Sentiment-based | Conference | 2021 | [161] |
| 36 | GSDNT | Sentiment-based | Journal (Elsevier) | 2021 | [162] |
| 37 | RF | Sentiment-based | Journal (Elsevier) | 2021 | [163] |
| 38 | NB | Sentiment-based | Conference | 2021 | [164] |
| 39 | CBM and BBM | Other approaches | Conference | 2020 | [165] |
| 40 | AT | Other approaches | Journal (Elsevier) | 2020 | [166] |
| 41 | LDA | Other approaches | Conference | 2020 | [167] |
| 42 | SVM, LR and PNN | Other approaches | Journal (Elsevier) | 2021 | [168] |
| 43 | REB | Other approaches | Journal (Elsevier) | 2021 | [169] |
| 44 | CM | Other approaches | Conference | 2021 | [170] |
| 45 | - | Other approaches | Conference | 2021 | [171] |
| 46 | EI | Other approaches | Journal (Elsevier) | 2021 | [172] |
| 47 | SWOT | Other approaches | Journal (MDPI) | 2021 | [173] |
| 48 | (GAN | Other approaches | Journal | 2021 | [174] |
| 49 | SVM and k-NN | Other approaches | Conference | 2021 | [175] |

## 5. Conclusions

This survey addresses several aspects of the spam reviews detection field. First by outlining all the previous surveys in this domain and their style of reviewing the works of spam reviews detection. In the second aspect, a background description of the spam reviews detection concepts is presented. While, the third aspect consists of analyzing and examining the works of spam reviews detection during the years 2020 and 2021, which corresponds to the COVID-19 period. Then a categorization of these works is divided into nine parts that their limitations, advantages, and how to improve them are explained. Furthermore, a comparison of the outcomes of our analysis and the works prior to the COVID-19 pandemic are also presented. For future work, a new survey should be presented that covers the spam reviews detection articles for three-period, before, during, and after the COVID-19 situation. In this way, all aspects of spam reviews and their detection evolution and development can be fully investigated.

**Author Contributions:** Conceptualization, A.M.A.-Z.; Writing original draft preparation, A.M.A.-Z.; Writing review and editing, A.M.A.-Z., A.M.M. and H.F.; Supervision A.M.M. and H.F. All authors have read and agreed to the published version of the manuscript.

**Funding:** This work has been partially funded by projects PID2020-113462RB-I00 (ANIMALICOS), granted by Ministerio Español de Economía y Competitividad; projects P18-RT-4830 and A-TIC-608-UGR20 granted by Junta de Andalucía, and project B-TIC-402-UGR18 (FEDER and Junta de Andalucía).

**Institutional Review Board Statement:** Not applicable.

**Informed Consent Statement:** Not applicable.

**Data Availability Statement:** No new data were created or analyzed in this study.

**Conflicts of Interest:** The authors declare no conflict of interest.

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
