# Peer review of "Spam Reviews Detection in the Time of COVID-19 Pandemic: Background, Definitions, Methods and Literature Analysis"

_applsci, doi:10.3390/app12073634_

Round 1

Reviewer 1 Report

The paper presents a literature analysis about spam reviews detection during the COVID-19 pandemic period. 

The paper is well written and structured. 

I have some doubts/suggestions:

1) I cannot understand where the database that the authors searched to select the works presented. It could be presented in the introduction, before talking about the types of approaches (figure 1). Because it is not a systematic literature review (there is not a methodology used), I want to know how these works were selected (inclusion criteria). 

2) Why presented in detail (figure 4 and 5) just one study?  Or the publishers and types are from this paper? I really cannot understand: "The publishers and type of studies of the reviewed works are summarized in Figures 4 and 5."

3) A table to summarize the advantages and disadvantages described in section 4 should be done. Three columns: approach, advantages, and disadvantages.

4) The discussion about "future directions" is superficial. It could be improved.

Typos questions:
1) pendemia -> pandemia
2) literate-> literature?
3) "Most of them focus on solving spam reviews using different detection systems, such as .." ->  incomplete statement?
4) therefore (in section 4) -> Therefore

Reviewer 2 Report

The authors carry the survey work on  "Spam Reviews Detection in the Time of COVID-19 Pandemic". The paper has potential but needs some issues to be fixed before acceptance. 

Grammatical issues:

-Mistake in the sentence: "As such, its import to detect spam reviews"
-Mistake in the sentence: "Most of them focus on solving spam reviews using different detection
 systems, such as ..
For instance, [156] implemented their detection system through a collective technique to
identify spammers and spam reviews (adversarial type).

Major issues:
-How did the authors collect all of those articles?
-It needs a clear background. During Covid-19, people are not only busy writing reviews but also active on social media. They are writing tweets/posts which could be important clues for sentiment analysis. It is good to include such things in the paper. For example, https://www.hindawi.com/journals/cin/2021/2158184/ and https://www.hindawi.com/journals/cin/2022/5681574/ could provide such information in the paper.
-The bibliometric analysis could provide us with important information
-The word cloud of keywords used in spam could also provide us with important information.
-Addition of sample snippets of fake or normal reviews here could be meaningful

Round 2

Reviewer 2 Report

I would like to thank the authors for their hard work to improve the manuscript. Given that manuscript has been revised carefully, the reviewer is inclined to accept. It still needs English proofreading in the revised version. Please revise them carefully. For eg., you may combine sentences 1 and 2 to make it clear and fix the grammatical issues. 

Author Response

Comment: I would like to thank the authors for their hard work to improve the manuscript. Given that manuscript has been revised carefully, the reviewer is inclined to accept. It still needs English proofreading in the revised version. Please revise them carefully. For eg., you may combine sentences 1 and 2 to make it clear and fix the grammatical issues.

Response: Thank you for your kind comment. The manuscript has been carefully reviewed and checked for any typographical errors or grammatical mistakes.